# General Characteristics of Children with Single- and Co-Infections and Febrile Seizures with a Main Focus on Respiratory Pathogens: Preliminary Results

**DOI:** 10.3390/pathogens10081061

**Published:** 2021-08-20

**Authors:** Henriette Rudolph, Katharina Gress, Christel Weiss, Horst Schroten, Ortwin Adams, Tobias Tenenbaum

**Affiliations:** 1Paediatric Infectious Diseases, University Children’s Hospital Mannheim, Medical Faculty Mannheim, Heidelberg University, 68167 Mannheim, Germany; henriette.rudolph@kgu.de (H.R.); katharina.gress@umm.de (K.G.); horst.schroten@umm.de (H.S.); 2Institute of Medical Statistics and Biomathematics, Medical Faculty Mannheim, Heidelberg University, 68167 Mannheim, Germany; christel.weiss@medma.uni-heidelberg.de; 3Institute of Virology, University Children’s Hospital, Heinrich-Heine-University, 40225 Düsseldorf, Germany; ortwin.adams@uni-duesseldorf.de

**Keywords:** febrile seizure, respiratory pathogens, HHV6, influenza, AV, rhinovirus, viral load

## Abstract

Febrile seizures (FS) affect up to 5% of children. The pathogen etiology in regard of viral loads has never been investigated. In a prospective cohort study we investigated the correlation between virus type and quantity in nasopharyngeal aspirates (NPAs) and the clinical characteristics in pediatric patients with a FS. From January 2014 to April 2016, 184 children with a FS were prospectively enrolled. The mean age of all included children was 26.7 ± 18.3 months with a male to female ratio of 1.4:1. Males with an acute disease and a short duration or absence of prior symptoms had a higher risk for complex FS. The majority of patients with FS presented with a generalized convulsion (180; 98%) and was admitted to hospital (178; 97%). Overall, 79 (43%) single and in 59 (32%) co-infections were detected. Human herpes virus 6 (HHV6), influenza, adenovirus (AV) and rhinovirus (RV) were the dominant pathogens, all detected with clinically significant high viral loads. HHV6 positive cases were significantly younger and less likely to have a positive family/personal history for FS. Influenza positives showed a higher rate of complex seizures, lower leukocyte and higher monocyte counts. AV positive cases were more likely to have a positive family history for FS and showed higher C-reactive protein values. In conclusion, a high viral load may contribute to the development of a FS in respiratory tract infections.

## 1. Introduction

Between 6 months and 5 years, febrile seizures (FS) affect 2–5% and are the most common neurologic disorder in children in Western Europe and North America [1]. The known risk factors include fever, familiar predisposition and infections [2].

Children with febrile seizures frequently present as fever of unknown origin (FUO). Therefore, a broad differential diagnosis needs to be considered. The most common cause of FUOs are infections with 37% to 59.3% depending on geographic, climatic, zoonotic and social factors. Non-infectious inflammatory diseases such as rheuma are the second differential diagnosis of FUO with around 14%, whereas malignancies are a rare but important cause of FUO [3,4,5]. Recently, FDG-PET/CT in children with FUO has gained in importance [6].

Already in the 1990s, an association of FS and viral infections at around 50% was described [7]. With the aid of modern detection methods for viral infections and rapid testing, the estimated rate of viral infection as fever source in FS has risen, but exact data in regard to viral loads is lacking [1,8]. Seasonal distribution of FS in line with regional endemics of influenza and enterovirus were described [5]. Additionally, upper respiratory tract infections were documented in around 50% of children with FS [1,9,10,11]. The second dominant diagnosis in children with FS was gastroenteritis, at up to 25% [10,11]. Modern molecular techniques have helped to generate evidence for the involvement of neurotropic viruses in the pathogenesis of FS [2].

Hence, this prospective single center study aims to fill the gap in knowledge regarding the association of respiratory viral infections and FS using both a qualitative and quantitative approach with qPCR.

## 2. Results

### 2.1. General Patient Characteristics

During the study period 184 children with FS were prospectively enrolled. The mean age of all included children was 26.7 ± 18.3 months (minimum 7, median 23 and maximum 118 months). The male to female ratio was 1.4:1. The mean temperature at admission before the onset of the FS was 39.0 ± 1.0 °C. The majority presented with a generalized convulsion (180; 98%). The rate of simple FS was 151 (82%). A complex seizure was detected in 33 (18%) of the patients. Males, with 26 affected patients, had a significantly higher rate/risk of complex seizures (3.3% higher) than females, with six affected patients (*p* = 0.0022). In 35 (19%) the use of benzodiazepines was required. In total, 178 (97%) of children were admitted to hospital. Data on the duration of symptoms before onset of FS were available for all patients. The mean duration of symptoms was 2.1 ± 4.0 days (mean: 1 day, range 0–30 days). Duration of symptoms ≥1 day were defined as “symptoms before seizure”, and 71/106 of males and 48/78 females suffered from symptoms ≥1 day before onset of FS (Table A1). Only 12/33 males with complex FS and 1/7 females with complex FS had “symptoms before seizure”. This indicates that a short duration or absence of prior symptoms had a higher risk for complex FS. Parallel multiple logistic regression of “sex” and “symptoms before seizure” revealed significant *p* values (*p* = 0.0041 for “sex” (male vs. female), Odds ratio = 3.916; *p* = 0.0005 for “symptoms before seizure” (present vs. absent), Odds ratio = 0.238).

In 70 cases (38%) an upper respiratory tract infection was suspected at admission. The most common main diagnosis in the medical report, with 59 cases (32%), was a lower respiratory tract infection (LRTI), followed by an upper respiratory tract infection (URTI) with 36 cases (20%), including rhinitis, pharyngitis and tonsillitis. Flu-like symptoms were observed in 26 cases (14%), otitis media in 24 cases (13%) and gastroenteritis in 25 cases (14%). Other diagnoses were exanthema subitum and stomatitis aphthosa, both reported in four cases each (2%), and Hand-Foot-Mouth Disease in three patients (2%). Additionally, two patients suffered from pyelonephritis and one was recently vaccinated against Neisseria meningitidis B (Figure 1).

In total, 77 children underwent EEG diagnostics, which is not generally recommended for the diagnostic work-up of FS in Germany. In 5 of 77, focal abnormalities were detected. One of these five patients was classified in the subgroup no pathogen, three patients had a single infection (RSV, AV, hPIV and one had a co-infection of HHV6 and Salmonella typhi. Three children underwent a diagnostic lumbar puncture (one had a single infection with HBoV and two had a co-infection with HHV6: 1 with HHV6 and *Salmonella typhi*, 1 with CoV). All children could be dismissed in a good state of health.

### 2.2. Distribution of Respiratory Viruses in Patients with Febrile Seizures

The most common pathogens in children with FS were HHV6, influenza, AV and RV (Figure 2a). Predominantly influenza and HMPV occurred as viral single infection, whereas the other infections as human metapneumovirus (HMPV), human bocavirus (HBoV), enterovirus (EV), human coronavirus (CoV) and human parainfluenza virus (hPIV) occurred as co-infections ((Figure 2a). Mean viral loads for HHV6 were 2.72 × 10^3^ ± 1.09 × 10^4^ copies/mL, for HBoV with 9.99 × 10^5^ ± 2.38 × 10^6^ copies/mL and for RV with 7.90 × 10^4^ ± 1.69 × 10^5^ copies/mL were the lowest (Figure 2b). In contrast, mean viral loads for AV were 1.28 × 10^9^ ± 4.33 × 10^9^ copies/mL, and for hPIV 1.31 × 10^8^ ± 1.97 × 10^8^ copies/mL as the highest (Figure 2b). The mean viral loads for the other respiratory pathogens influenza, EV, CoV, RSV and HMPV were all above 2 × 10^6^ copies/mL (Figure 2b).

### 2.3. Comparison of Clinical Characteristics in Patients with Viral Single, Co-Infections, and Virus Negative Patients

In 138/184 children (75%) with FS a viral pathogen was identified within the nasopharyngeal aspirates (NPA; Figure A1). In 79 (43%) patients a single infection, and in 59 (32%) patients a co-infection was detected (Figure A1). Among the 79 patients with single infection stool samples, two patients were positive for norovirus, one for rotavirus and one patient suffered from pyelonephritis and Enterobacter was identified within the urine sample. Among the 59 patients with co-infections, one co-infection was with HHV6 and Salmonella typhi. In 63 patients, blood cultures were taken, and all cultures remained negative (data not shown).

Children with co-infections were 26.2 ± 15.8 months younger than children with single infections (29.3 ± 20.0) or no identified infections (30.9 ± 18.4) (Table 1). Between 86% (single infections) and 95% (co-infections) of patients suffered from fever before the onset of the convulsion. In regard to the following clinical characteristics, course of the seizure, hospital admission and family and personal history of FS or epilepsy, no significant differences were detected between patients with no pathogen, single infection, or co-infection (Table 1). Most of the study patients had a generalized simple FS.

Almost all of the study patients (178; 97%) were admitted to hospital. Children with co-infections received benzodiazepines, in 18/59 (31%) of cases, significantly more often than children with single infections (10/184; 13% *p* = 0.0099; Table 1). As differences between single and co-infections were not unequivocal, further subgroup analysis with regard to the underlying pathogen was performed.

### 2.4. Clinical and Laboratory Characteristics of the Four Most Frequently Detected Pathogens with Febrile Seizures

Taking a closer look at the four most prevalent viral pathogens, namely HHV6, influenza, AV and RV, and comparing single and co-infections, important differences in the clinical and laboratory parameters within these eight groups, compared to the patients without a pathogen, could be identified (Table 2). Children with HHV6 single or co-infection were, with 18.1 ± 4.8 and 26.7 ± 17.5 months, significantly younger than patients with no pathogen, with 30.9 ± 18.4 months (*p* = 0.0382; Table 2). Moreover, they were, with 1/15 (7%) cases, less likely to have a positive family history of FS than children with HHV6 co-infections (15/40; 38%; *p* = 0.0238; Table 2). This was in contrast to infections with AV, in which AV single infection was more often associated with a positive family history for FS than AV co-infection (*p* = 0.0419; Table 2). Interestingly, children with AV, influenza and RV single infections had, with four (36%), 5 (20%) and two (40%) cases among AV, influenza and RV single infections, the highest proportion of complex convulsions (Table 2; Table A2). Moreover, children with FS and an influenza co-infection received significantly more benzodiazepines than children with influenza single infection (4/14 (29%) vs. 1/25 (4%); *p* = 0.0469; Table 2). None of the children with influenza infection had received an influenza vaccination earlier.

Concerning the laboratory parameters, typical features of the specific viral pathogen in single or co-infection were documented. Patients with influenza single infection displayed a significant lower leukocyte count of 7.7 × 10^3^/µL than patients without a viral infection (*p* = 0.0185; Table 2), whereas monocytes were significantly higher with 10.7% compared to 6.7% in patients without a viral pathogen (*p* = 0.0185; Table 2). AV single infections had a significantly higher leukocyte count of 7.7 × 10^3^/µL than in children without a viral pathogen detection (*p* = 0.0010; Table 2). Moreover, both AV single and co-infections had a significant higher CrP at admission with 20.4 mg/L (*p* = 0.0091) in AV single infections and 16.4 mg/L (*p* = 0.0186) in AV co-infections compared to the virus negative group (Table 2). The maximum CrP reached in AV infection was also significantly higher both in AV single infection (27.1 mg/L, *p* = 0.0044) and AV co-infection (19.3 mg/L; *p* = 0.0078) than in patients without a virus detected (Table 2).

## 3. Discussion

The main aim of this prospective study was to determine the dominant pathogens in children with FS and assess their viral loads. We have further analyzed whether there was a specific pattern of clinical and laboratory findings for the respective pathogens.

The major strengths of this pediatric clinical study were the prospective design over two consecutive years and the application of a highly sensitive quantitative multiplex real-time PCR for a wide range of respiratory viruses in children presenting with FS. A limitation is the lack of an age-balanced control group with febrile infections and without FS during the study period. Moreover, recruitment was performed in the hospital so that FS that presented directly as outpatients at their local doctor were not included. Results and conclusions should be interpreted in this context with caution.

In our study, the male to female ratio was, at 1.4 to 1, similar to ratios described in the literature [12]. We detected with 82% of simple FS a comparable rate to the 80 to 85% described in the literature [1,13]. Interestingly, in a detailed logistic regression model the combination of the factors “male sex” and “absence of symptoms before FS” was associated with a significantly higher risk of the development of a complex FS. In other words, male children who presented with an acute disease course, represented by a duration of disease symptoms ≤1 day, had the highest risk for complex FS. So far it is reported that males have a slightly elevated risk for the development of FS [12], but data on the risk for males presenting with complex FS have not yet been published.

In 74% of cases a viral pathogen was detected, which is in line with a recently published Korean retrospective analysis of FS that found in 81.2% of the 607 cases of FS tested a positive result for respiratory viruses such as RV, EV, AV or influenza [13]. Before the wide application of routine PCR techniques for respiratory pathogens was introduced, the rate of viral infections in FS was estimated at approximately 50% [2,3].

The mean viral loads for all detected viruses within the NPA were within the range described for significant acute single viral infections in the literature. Viral loads for HHV6 were above the threshold of 1 × 10^3^ viral copies/mL that is considered for the detection of an active infection in whole blood [14]. The mean viral titers for detected influenza were 1.5 × 10^7^ viral copies/mL above the level of 1 × 10^6^ viral copies/mL documented within the NPA of children with respiratory tract infection [15].

In two large prospective studies in children, it was suggested, that RV loads above 10^4^ copies/mL are very likely to be the presenting cause of illness [16,17], suggesting that the patients included in our study showed clinically significant viral loads. As to the viral load of HBoV, a prospective study in children under five years of age found that, in healthy controls, the median viral load was 2.7 × 10^3^ copies/mL, whereas outpatients and inpatients with respiratory symptoms had higher median viral loads of 2 × 10^6^ and 5.1 × 10^6^ copies/mL, respectively [18]. In our study, the mean viral load was 9.99 × 10^5^ copies/mL. The viral loads for the other pathogens detected (influenza, AV, EV, CoV, RSV hPIV and HMPV) were all above the cut off values defined in the literature for symptomatic patients [19,20,21].

In our prospective study, HHV6, influenza, AV and RV were the dominant pathogens detected in children with FS, which is in line with a recent retrospective analysis on FS and viral infections [13,22]. HHV6 infection in children with FS is controversially discussed within the literature [23,24]. In a large retrospective study on primary HHV6, infection was documented in 24% of patients with FS [8]. In a prospective observational study involving children with febrile illness, 9.7% of the children had primary HHV6 infection and 13% of these had a FS [25]. In a case control study that applied both qualitative viral detection and serological testing, the positive association of HHV6 primary infection and FS could be detected [26]. In contrast, in a more recent case control study on children with FS, age-matched febrile children and healthy children, HHV6B DNA was detected in 50% of patients with FS, compared to 37% in fever controls and 35% in healthy individuals [27]. However, in this data, no information on the detected viral loads was available. To date, HHV6 infection seems to play a role in the multifactorial genesis of FS, but future prospective case control studies including serological testing, which is lacking within our study, as well as quantitative PCR, are needed.

Influenza was detected as the second most frequently detected pathogen in our study. In this context it may be relevant to note that during both study seasons 2014/2015 and 2015/2016 the number of laboratory confirmed cases of influenza registered within the national surveillance in Germany was relatively high, with 80.600 cases for 2014/2015 and 71.100 cases in 2015/2016 [28,29]. Importantly, we observed a significantly high proportion of patients with influenza A infection who developed a complex FS. It is known that the most common neurologic manifestations of influenza in children are FS and that there is an increasing perception of other neurologic complications [30,31]. Still, so far both the risk for FS and the clinical course in influenza infection are not well investigated. A Chinese prospective study analyzed the special characteristics of FS in children with influenza A infection. Here, a high proportion of complex seizures compared to a control group was documented [32]. Especially in regard to public health recommendations such as vaccination, future studies should focus on this topic. As described, the patients with influenza single infection displayed lower leukocyte but higher monocyte counts compared to patients without a documented (viral) infection [33].

Interestingly, both seasons in which the study was performed had a high number of AV infections with a peak in patients below 2 years, estimated by the rate of AV detected within sentinel samples of influenza surveillance [28,29].

Recently, two large retrospective studies supported the role of AV infections in (complex) FS [34,35]. Among the CNS complications of AV infections, FS account for more than half of the cases [34]. We further detected the highest CrP values in patients suffering from AV infections, which is in congruence with the literature [36].

In contrast to previous studies [13] EV was detected at a relative low rate within our study cohort, which might be explained by the absence of major EV outbreaks during the study period as documented by the national enterovirus surveillance (https://evsurv.rki.de/Content/Query/Create.aspx, accessed 1 July 2021).

In conclusion, we could identify male sex in combination with a short duration or absence of symptoms prior to FS as risk factors for the presence of a complex FS. Moreover, we detected in our prospective cohort study in children with FS a variety of respiratory viruses with a clinically significant viral load. HHV6, influenza, AV and RV were the dominant viral pathogens in our patient cohort. Interestingly, FS in patients with influenza single and co-infections led to a higher benzodiazepine use. Single infections with AV, influenza and RV were associated with a higher rate of complex FS than respective co-infections. Future studies on influenza infections and CNS complications are needed, especially in regard to general recommendations for influenza vaccination.

## 4. Materials and Methods

### 4.1. Study Population and Data Acquisition

A prospective study was performed from January 2014 to April 2016 at the University Children’s Hospital, Mannheim, Heidelberg University. Identification and recruitment of patients was made by the responsible physician in the emergency department or the unit within the hospital after admission. We included children between 0 and 18 years presenting with a FS after informed consent of the parents was obtained. FS were defined as seizures that occurred in children in the context of fever without a (pre-)existing epilepsy or CNS infection. Consequently, children with epilepsy or CNS infection were excluded from the study. Inclusion criteria were age between 0 and 18 years, minimum 1 FS and written informed consent/assent for study participation.

NPAs and clinical data were prospectively collected using a standard data collection form. Additional diagnostics such as chest radiogram, blood sampling for laboratory parameters or blood cultures, analysis of the urine or lumbar puncture were not a study requirement. Results of additional diagnostics were documented and analyzed within the study when available.

### 4.2. Analysis of Nasopharyngeal Aspirates

On the day of admission, NPAs were collected and evaluated in the Institute of Virology in Düsseldorf with q-PCR for HHV6, AV, RSV A and B, RV, CoV, Influenza A and B, Influenza A H1N1 (H1N1), parainfluenza 1, 2 and 3 (hPIV), EV (coxsackie A and B viruses and echoviruses except for polioviruses 1–3), HboV and HMPV. The quantification of the viruses was performed by a previously described 1-step real-time PCR method of which sensitivity and specificity have been demonstrated elsewhere [11,37,38].

In brief, the Quantitect Multiplex q-PCR kit (No.204643, Qiagen, Hilden, Germany) containing HotStarTaq DNA Polymerase was used for the assay. Amplification was performed in an ABI7500 thermocycler using conditions as recommended by the manufacturer of the multiplex RT-PCR kit. As standards, DNA plasmids that encompass the amplified region were created and serially diluted after purification. Standard graphs of the CT values obtained from serial dilutions of the standards were constructed and the numbers of specific genomes were calculated by the software. NPA as a method for the detection of a viral infection was chosen as a non- invasive procedure, for which reference values for the viral load in copies per milliliter (mL) of respiratory viruses exist [11].

Following the symptoms, additional cultures or urine and/or stool and blood cultures were performed and analysed within the department of microbiology of the University Hospital Mannheim with the established routine tests. Virus testing of stool samples was performed by real time PCR with Cepheid SmartCycler^®^ (Cepheid GmbH, Krefeld, Germany) with the following kits: Smart Norovirus (Genzyme Virotech GmbH, Rüsselsheim, Germany) for Norovirus and RIDA^®^GENE Viral Stool Panel II (R-Biopharm AG, Darmstadt, Germany) for rotavirus, AV und astrovirus.

### 4.3. Statistics

Qualitative factors are presented as absolute and relative frequencies, and quantitative parameters are given as mean and standard deviation or median and range. Clinical characteristics and laboratory variables, approximately normally distributed, have been compared using, as univariable analysis, 2 sample t tests, with the pooled method in the case of similar variances or the Satterthwaite method otherwise. For skewed distributed quantitative data, Mann–Whitney U test has been performed. To compare frequencies of 2 samples, χ^2^ test or Fisher’s exact test has been used; for ordinal scaled data, Cochran–Armitage trend test was preferred. A test result with a 2-sided *p*-value < 0.05 was considered statistically significant. For significant test results, Odds ratios were calculated. Additionally, multiple logistic regression was performed using the “selection = stepwise” option. For the multiple analysis, all variables have been chosen with *p* < 0.10 in the univariable analyses. All statistical calculations have been done with the SAS system, release 9.4 (SAS Institute Inc., Cary, NC, USA).

### 4.4. Ethical Statement

Approval was provided by the local ethics committees (Medical Faculty of Mannheim, Heidelberg University [2013-643N-MA]. Written informed consent was obtained from parents along with assent from children (when possible, according to their age) before any study procedures being performed.

## Figures and Tables

**Figure 1 pathogens-10-01061-f001:**
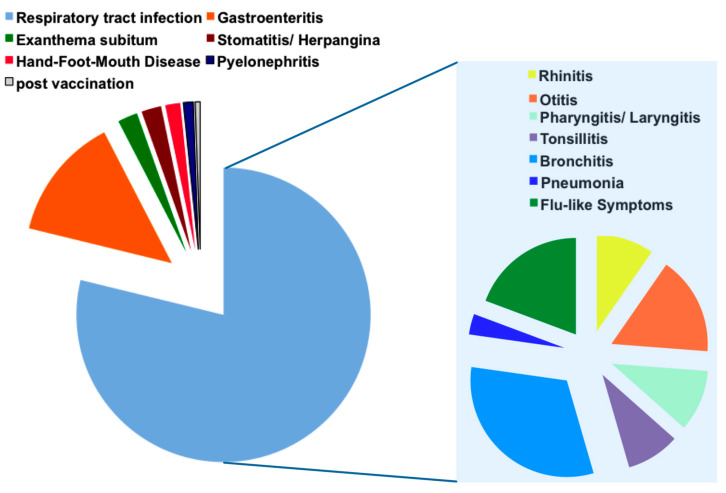
Main diagnosis of children with febrile seizures; on the left, distribution of main diagnosis in children with febrile seizures are listed; on the right, the diagnosis “respiratory tract infection” is further specified; absolute numbers of cases are visualized.

**Figure 2 pathogens-10-01061-f002:**
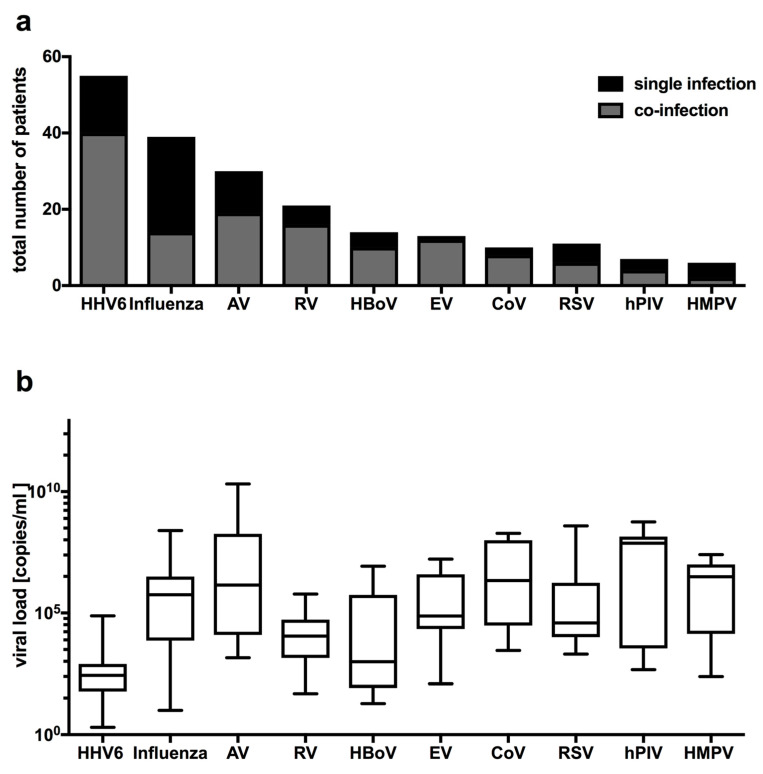
Distribution of viral pathogens in children with febrile seizures: (**a**) single (black) and co-infections (grey) displayed as total number of patients; (**b**) viral load in children with febrile seizures.

**Table 1 pathogens-10-01061-t001:** Clinical characteristics of children with febrile seizures without or with a viral single or co-infection.

	No Pathogen	Single Infection	Co-Infection	*p*-Value *
**Total number**	46	79	59	
**Age (months)**	30.9 ± 18.4	29.3 ± 20.0	26.2 ± 15.8	0.3150
**Sex**				
Male	25 (54%)	48 (61%)	33 (56%)	0.5688
Female	21 (46%)	31 (39%)	26 (44%)	
**Fever before seizure**	41 (89%)	68 (86%)	56 (95%)	0.0889
**Temperature during seizure**	39.1 ± 0.9 °C	38.9 ± 1.1 °C	39.2 ± 0.9 °C	0.0734
**Type of seizure**				
Focal	2 (4%)	2 (3%)	0 (0%)	0.5069
Generalized	44 (96%)	76 (97%)	60 (100%)	
Simple	37 (80%)	65 (82%)	49 (83%)	0.9057
Complex	9 (20%)	14 (18%)	10 (17%)	
**Cyanosis**	14 (30%)	20 (25%)	13 (22%)	0.6547
**Hospital admission**	46 (100%)	75 (95%)	57 (97%)	1.0000
**Benzo-diazepine use**	7 (15%)	**10 (13%)**	**18 (31%)**	**0.0099**
**Family history**				
Febrile seizures	12 (26%)	15 (19%)	18 (31%)	0.1165
Epilepsy	4 (9%)	14 (18%)	4 (7%)	0.0590
**Personal history**				
Febrile seizures	4 (9%)	5 (6%)	8 (14%)	0.1503
Epilepsy	1 (2%)	0 (0%)	1 (2%)	0.4275

Absolute numbers and, in parenthesis, percentages for the listed characteristics are listed; * *p* calculated between the following two groups: pathogen single vs. co-infection.

**Table 2 pathogens-10-01061-t002:** Clinical characteristics of children with febrile seizures in the sub-groups with Human herpes virus 6 (HHV6), influenza, Adenovirus (AV) or Rhinovirus (RV) single or co-infections.

	**No** **Pathogen**	**HHV6** **Single**	**HHV6** **Co**	***p*-Value**	**Influenza** **Single**	**Influenza** **Co**	***p*-Value**	**AV** **Single**	**AV** **Co**	***p*-Value**	**RV** **Single**	**RV** **co**	***p*-Value**
**Total patient number**	46	15	40		25	14		11	19		5	16	
**Age (months)**	30.9 ± 18.4	18.1 ± 4.8	26.7 ± 17.5	**0.0382**	42.4 ± 26.1	32.0 ± 24.1	0.1021	27.5 ± 12.5	25.5 ± 12.1	0.4551	19.6 ± 7.6	25.9 ± 10.1	0.2468
**Complex** **seizure**	9 (20%)	1 (7%)	7 (18%)	0.5052	5 (20%)	2 (14%)	0.0677	4 (36%)	4 (21%)	0.4928	2 (40%)	1 (6%)	0.1578
**Benzo-diazepine use**	8 (17%)	1 (7%)	11 (28%)	0.1485	**1 (4%)**	**4 (29%)**	0.0949	2 (18%)	6 (32%)	0.3503	1 (20%)	6 (37%)	0.1746
**Positive family** **history FS**	12 (26%)	**1 (7%)**	**15 (38%)**	0.0612	4 (16%)	2 (14%)	0.4809	6 (55%)	3 (16%)	0.0682	0 (0%)	6 (37%)	0.3540
**Positive** **personal** **history FS**	4 (9%)	1 (7%)	4 (10%)	1.0000	0 (0%)	1 (7%)	0.2979	0 (0%)	5 (26%)	0.0695	0 (0%)	4 (25%)	0.2676
**Leukocytes (µL)**	10.4 (7.1–12.4)	11.9 (9.7–13.6)	10.0 (9.2–14.9)	0.3821	**7.7** **(4.5–11.0)**	**10.0** **(8.4–13.7)**	**0.0273**	**15.4** **(13.8–17.8)**	**12.0** **(10.0–16.2)**	**0.0021**	11.0 (9.2–15.7)	9.6 (8.0–12.7)	0.8101
**Monocytes (%) ***	7.8 (4.4–10.0)	8.0 (5.0–10.0)	7.0 (4.0–9.7)	0.9766	**10.7** **(9.1–13.4)**	**6.7** **(2.5–9.4)**	**0.0304**	5.3 (1.5–9.2)	12.0 (6.0–12.4)	0.3547	-	9.0 (4.0–12.4)	0.6202
**CrP (at admission) (mg/L)**	10.8 (7.0–17.8)	15.5 (7.2–22.4)	12.8 (7.4–23.8)	0.6350	9.6 (5.9–16.5)	12.8 (4.8–21.3)	0.7153	**20.4** **(15.9–50.0)**	**16.4** **(12.1–29.8)**	**0.0067**	11.9	15.3 (9.3–45.5)	0.3719
**CrP (max.)** **(mg/L)**	11.3 (6.8–20.4)	18.0 (8.2–46.0)	18.0 (8.2–46.0)	0.4220	9.6 (5.9–16.4)	12.8 (4.8- 21.3)	0.7378	**27.1** **(16.8–72.4)**	**19.3** **(12.1–49.4)**	**0.0021**	11.9	19.3 (8.9- 62.3)	0.5137

Quantitative variables are presented by mean ± SD; for categorical factors, absolute and relative frequencies are given; laboratory. Parameters are presented by median together with interquartile range *p* calculated for the three groups: no pathogen vs. single infection vs. co-infection; in case of *p* ≤ 0.05, subgroup analysis between single and co-infection was performed and described within Section 2.4. * % of total leukocytes. CrP = C reactive protein.

## Data Availability

Data sharing not applicable.

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
