# Peer review of "General Characteristics of Children with Single- and Co-Infections and Febrile Seizures with a Main Focus on Respiratory Pathogens: Preliminary Results"

_pathogens, 2021, doi:10.3390/pathogens10081061_

Round 1

Reviewer 1 Report

I read carefully this manuscript. The idea of the manuscript is very good. However, I have comments and recommendations to this manuscript.   

1. Introduction: I recommend to this section to add 2-3 sentences for "Fever of Unknown Origin (FUO)" among children. In this regard, I recommend to add the following scientific publications to this section (and add to section "References"):

  • Chien YL, Huang FL, Huang CM, Chen PY. Clinical approach to fever of unknown origin in children. J Microbiol Immunol Infect. 2017; 50(6): 893-898. [DOI: 10.1016/j.jmii.2015.08.007] [PMID: 27143687].   
  • Baymakova M, Plochev K, Dikov I, Popov GT, Mihaylova-Garnizova R, Kovaleva V, et al. Fever of unknown origin in a Bulgarian hospital: Evaluation of 54 cases for a four year-period. J Clin Anal Med. 2016; 7(1): 70-75. [DOI: 10.4328/JCAM.3897].
  • Pijl JP, Kwee TC, Legger GE, Peters HJH, Armbrust W, Schölvinck EH, Glaudemans AWJM. Role of FDG-PET/CT in children with fever of unknown origin. Eur J Nucl Med Mol Imaging. 2020; 47(6): 1596-1604. [DOI: 10.1007/s00259-020-04707-z] [PMID: 32030452].
  • Szymanski AM, Clifford H, Ronis T. Fever of unknown origin: A retrospective review of pediatric patients from an urban, tertiary care center in Washington, DC. World J Pediatr. 2020; 16(2): 177-184. [DOI: 10.1007/s12519-019-00237-3] [PMID: 30888665].

2. Materials & Methods: I recommend, to add "Case Definition" for "Febrile Seizure (FS)."

3. Materials & Methods: I recommend, to differentiate separate subsection "Inclusion and Exclusion Criteria" (4-5 sentences).

4. Results, Table 1: I recommend, to change the term "gender" with a new term "sex."

5. Results: I think that main limitation of your research is the lack of clinical characteristics (detailed medical history in each clinical case; laboratory results; results of imaging studies, etc.). I think that this seriously reduces scientific qualities of this manuscript. Essentially, this manuscript has a descriptive character and gives just a "general picture" of the medical problem under consideration. I will be happy to listen your answers for these serious limitations of your research.

6. Results: I saw and one more serious limitation of your research: almost missing statistical analysis. You have calculated only "P-value" and "Mean±SD". They are not calculated important statistical indicators such as: Odds Ratio (OR), 95% Confidence Interval (95% CI), Parameter Estimate (PE), Standard Error (SE), etc. Not performed Univariate Analysis and Multivariate Analysis. That's why, I claim that this manuscript there is a very weak statistical analysis. I will be happy to listen your answers for this serious limitation.

7. Discussion: I recommend at the end of this section to add new paragraph "Limitations of the Study" (3-4 sentences). I recommend the last sentence in this new paragraph to be the following: "In this regard, the results and conclusions should be interpreted with caution.".

8. Title: In a relationship with the limitations I mentioned I recommend to change the title of the manuscript with the following new title: "General Characteristics of Children with Single-and Co-Infections and Febrile Seizures with a Main Focus on Respiratory Pathogens: Preliminary Results".  

Author Response

Please find enclosed our manuscript “Characteristics of Children with single- and co-infections and febrile seizures with a focus on respiratory pathogens” that we would like to submit for publication as an Research-Article in the “Pathogens” as part of the special issue “Respiratory Tract Infections in Children”.

In the revised version of our manuscript the concerns and suggestions of the reviewers have all been addressed and changes are marked up using the “Track Changes” function. Additionally, we updated the references. We thank all reviewers for their time, their suggestions, thoughts and valuable critique. With the help of this critical input the revised manuscript has significantly improved.

Please find below our point-by-point reply to each comment of the Reviewers.

Reviewer 1

1. Introduction: I recommend to this section to add 2-3 sentences for "Fever of Unknown Origin (FUO)" among children. In this regard, I recommend to add the following scientific publications to this section (and add to section "References"):

  • Chien YL, Huang FL, Huang CM, Chen PY. Clinical approach to fever of unknown origin in children. J Microbiol Immunol Infect. 2017; 50(6): 893-898. [DOI: 10.1016/j.jmii.2015.08.007] [PMID: 27143687].   
  • Baymakova M, Plochev K, Dikov I, Popov GT, Mihaylova-Garnizova R, Kovaleva V, et al. Fever of unknown origin in a Bulgarian hospital: Evaluation of 54 cases for a four year-period. J Clin Anal Med. 2016; 7(1): 70-75. [DOI: 10.4328/JCAM.3897].
  • Pijl JP, Kwee TC, Legger GE, Peters HJH, Armbrust W, Schölvinck EH, Glaudemans AWJM. Role of FDG-PET/CT in children with fever of unknown origin. Eur J Nucl Med Mol Imaging. 2020; 47(6): 1596-1604. [DOI: 10.1007/s00259-020-04707-z] [PMID: 32030452].
  • Szymanski AM, Clifford H, Ronis T. Fever of unknown origin: A retrospective review of pediatric patients from an urban, tertiary care center in Washington, DC. World J Pediatr. 2020; 16(2): 177-184. [DOI: 10.1007/s12519-019-00237-3] [PMID: 30888665].

Author's Response: We thank the Reviewer for this suggestion. Still we found it difficult to expand the introduction towards the field of "Fever of Unknown Origin (FUO)" among children as to completely picture this very interesting field, we would need to significantly expand the introduction. Additionally, in the literature available on febrile seizures, the topic “FUO” has never been of a point of discussion and febrile seizure is not mentioned as symptom in the one of the publications suggested by the Reviewer. That was why we decided to leave the focus on “febrile seizure” instead. If a considerable expansion of the introduction is requested, of course we could include additional studies on FUO.

2. Materials & Methods: I recommend, to add "Case Definition" for "Febrile Seizure (FS)."

Author's Response: We added the definition of Febrile Seizure to the Materials and Methods section under 4.1.

3. Materials & Methods: I recommend, to differentiate separate subsection "Inclusion and Exclusion Criteria" (4-5 sentences).

Author's Response: We added in- and exclusion criteria to section 4.1.

4. Results, Table 1: I recommend, to change the term "gender" with a new term "sex."

Author's Response: According to the reviewer`s recommendation we introduced the term “sex”.

5. Results: I think that main limitation of your research is the lack of clinical characteristics (detailed medical history in each clinical case; laboratory results; results of imaging studies, etc.). I think that this seriously reduces scientific qualities of this manuscript. Essentially, this manuscript has a descriptive character and gives just a "general picture" of the medical problem under consideration. I will be happy to listen your answers for these serious limitations of your research.

Author's Response: We thank the reviewer for bringing up this issue. When preparing the manuscript, we discussed if we should include all the details on medical history, laboratory results etc. for each case. Still, for 184 children, this would be rather difficult. That was why we grouped the data of the children with FS first in three groups: children without a detected pathogen, children with a single infection and children with co-infection (2 or more pathogens). We included the clinical data on these three groups in Table 1. For Table 1, we decided not to introduce the results of laboratory testing, as no specific differences could be detected. Next, we picked up the four dominant pathogens in our study population namely HHV6, Influenza, AV and compared each in single and in co-infection to the group „no pathogen“. In Table 2 you can find the detailed information about the clinical characteristics as age, seizure type, benzodiazepine use, personal/ family history as well as the laboratory characteristics at admission.

We added now some data on EEG diagnostics and lumbar puncture to section 2.1. Moreover, we added a supplemental table on the differences in “symptoms before seizure” and “type of seizure” (simple vs. complex) between males and females and some additional data to section 2.1.

6. Results: I saw and one more serious limitation of your research: almost missing statistical analysis. You have calculated only "P-value" and "Mean±SD". They are not calculated important statistical indicators such as: Odds Ratio (OR), 95% Confidence Interval (95% CI), Parameter Estimate (PE), Standard Error (SE), etc. Not performed Univariate Analysis and Multivariate Analysis. That's why, I claim that this manuscript there is a very weak statistical analysis. I will be happy to listen your answers for this serious limitation. 

Author's Response: We thank the Reviewer for this comment as it showed us, that the description of the section statistical analysis in the Material and Methods sections needs to be improved in order to point out, that thorough statistical analysis of all included data was performed with the help of our local department of statistics. As described in section 4.3. „Qualitative factors are presented as absolute and relative frequencies, quantitative parameters are given as mean and standard deviation or median and range“. Laboratory parameters for example were displayed as range.

Additional statistical analysis as Spearman's rank correlation coefficient were also performed. We updated the section 4.3.

Following the comment of Reviewer 1, we discussed with our statistician about additional analysis also in regards of the results we already documented in section 2. We decided to focus on the characteristic “complex seizure” and tested various possible factors of influence. For significant test results, Odds ratios were calculated and multiple logistic regression was performed using the "selection = stepwise" option. For the multiple analysis, all variables have been chosen with p < 0.10 in the univariable analyses. Interestingly, the detailed analysis revealed that the male sex and a short or absent presence of symptoms before the onset of the febrile seizure was associated with a higher risk for a complex febrile seizure.

The additional significant results were added to section 2.1 and 3.0. within the third paragraph. We think with this analysis we could significantly improve our manuscript. In addition, we rechecked with 2 independent persons again all calculations performed and all numbers displayed within text, figures and tables of the manuscript.

7. Discussion: I recommend at the end of this section to add new paragraph "Limitations of the Study" (3-4 sentences). I recommend the last sentence in this new paragraph to be the following: "In this regard, the results and conclusions should be interpreted with caution."

Author's Response: In the second paragraph of the discussion we already introduced what we found as one major limitation of the study namely the lack of an „age-balanced control group with febrile infections without FS during the study period“. We expanded this section. As suggested by the Reviewer we added the information "In this regard, the results and conclusions should be interpreted with caution."

8. Title: In a relationship with the limitations I mentioned I recommend to change the title of the manuscript with the following new title: "General Characteristics of Children with Single-and Co-Infections and Febrile Seizures with a Main Focus on Respiratory Pathogens: Preliminary Results".  

Author's Response: Following the Reviewers valuable comments, we tried to improve the manuscript and added more information as during the study period of two years thorough documentation of all clinical characteristics of the 184 patients included in this prospective analysis was performed. Since we plan no further follow-up, the data presented are final and not preliminary. Therefore, we changed the title to “Characteristics of children with single- and co-infections and febrile seizures with a main focus on respiratory pathogens”.

We hope to have adequately addressed all issues raised by the reviewers and do hope our revised manuscript is now acceptable for publication in Pathogens.

Yours sincerely,

Prof. Dr. Tobias Tenenbaum

Reviewer 2 Report

I have read this paper with great interest, and consider the data as reported as relevant and contributing to this clinical ‘syndrome’. I have provided some comments, with the request to the authors to consider these, and reflect on them, and if relevant, revise the paper.

A febrile seizure refers to an event in infancy or childhood, usually occurring between six months and five years of age, associated with fever but without evidence of intracranial infection or defined cause [Berg et al, Epilepsia 2010]. However, cases up to 18 year have been included. Can the authors comment or reflect on this?

All cases have been hospitalized and recruitment was at the ER: although pragmatic in its design, to what extent can this result in biases, as febrile convulsions are in themselves not an indication for hospitalization? Is the current cohort ‘representative’ for the population?

The age related pattern, with HHV6 positivity more common in the younger age and FS. To what extent is this specific to FS or is this ‘just’ a reflection of the epidemiology of the pathogen, or to rephrase this, is this different from non FS epidemiology?

You were able to confirm that the viral load (nasal swab) is indeed high, but not sure if this links causal to the FS event(s), rather association in my opinion. Can you comment on this?

Results: I assume that ‘co-infections’, or ‘non-pathogen‘ are restricted to the viral panel explored ?

Ethics: Technically, children can give assent, no formal legal consent, so suggest to rephrase.

Editing Comments: Suggest to check if each legend needs information on the abbreviations used (you could check some other papers in the journal); Do we need the ‘bullet’ points in table 1?

Author Response

Please find enclosed our manuscript “Characteristics of children with single- and co-infections and febrile seizures with a focus on respiratory pathogens” that we would like to submit for publication as an Research Article in Pathogens as part of the special issue “Respiratory Tract Infections in Children”.

In the revised version of our manuscript the concerns and suggestions of the reviewers have all been addressed and changes are marked up using the “Track Changes” function. Additionally, we updated the references. We thank all reviewers for their time, their suggestions, thoughts and valuable critique. With the help of this critical input the revised manuscript has significantly improved.

Please find below our point-by-point reply to each comment of the Reviewers.

Reviewer 2

I have read this paper with great interest, and consider the data as reported as relevant and contributing to this clinical ‘syndrome’. I have provided some comments, with the request to the authors to consider these, and reflect on them, and if relevant, revise the paper.

A febrile seizure refers to an event in infancy or childhood, usually occurring between six months and five years of age, associated with fever but without evidence of intracranial infection or defined cause [Berg et al, Epilepsia 2010]. However, cases up to 18 year have been included. Can the authors comment or reflect on this?

Author's Response: The reviewer is right that FS usually occur between six months and five years. As it remains a bit controversial within the literature, if up to 5 percent of FS persist beyond the age (Gencpinar, P., et al, 2017. The risk of subsequent epilepsy in children with febrile seizure after 5 years of age. Seizure: European Journal of Epilepsy 53, 62–65. doi:10.1016/j.seizure.2017.11.005) we decided to expand our age group for patient recruitment in order to have a more complete picture. In fact, the youngest patient included was 7 months old and the 3 patients older than 5 years (1 patient was 7 years old, the other 2 9 years old) all had an infection with Influenza A (1 in co-infection with HHV-6). We now added this information on Minimum, Median and Maximum to section 2.1.

All cases have been hospitalized and recruitment was at the ER: Although pragmatic in its design, to what extent can this result in biases, as febrile convulsions are in themselves not an indication for hospitalization? Is the current cohort ‘representative’ for the population?

Author's Response: As the Reviewer observed correctly recruitment was performed at the ER. Cases presented directly at the local doctor in charge as outpatients could not be included. We added this information to section 3 as limitation of the study. Still, from continuous exchange with our local pediatricians in Mannheim, Germany we perceived that the majority of patients with FS will be addressed to the hospital for evaluation.

The age related pattern, with HHV6 positivity more common in the younger age and FS. To what extent is this specific to FS or is this ‘just’ a reflection of the epidemiology of the pathogen, or to rephrase this, is this different from non FS epidemiology?

Author's Response: As disputed in section 3, the involvement of HHV6 in the pathogenesis of FS is discussed controversially within the literature. A reason could be that studies did not always include quantitative measurements of viral loads to define the cases. Also in our study, due to the fact that blood samples beyond the routine testing that would be performed in non-study patients was not permitted from the ethical point of view, serological testing for HHV-6 is lacking. This is a major limitation in data interpretation which we added in the discussion section in paragraph 2 and 6.

You were able to confirm that the viral load (nasal swab) is indeed high, but not sure if this links causal to the FS event(s), rather association in my opinion. Can you comment on this?

Author's Response: We totally agree with the Reviewer that detection of (significant) viral-load is rather an association that clear cause. That is why we already wrote in the Introduction “the association of respiratory viral infections and FS using both a qualitative as quantitative approach with qPCR”. In the conclusion we rephrased in order to keep this interpretation clear that it is an association rather than clear cause.

Results: I assume that ‘co-infections’, or ‘non-pathogen‘ are restricted to the viral panel explored?

Author's Response: We thank the Reviewer for this question and try to make it more clear as we cannot agree completely.

Additional screening despite the viral PCR was not mandatory for the study patients. As stated in section 4.2 “following the symptoms, additional cultures or urine and/ or stool and blood cultures were performed and analysed within the department of microbiology of the University....” Consequently, in 63 of 184 patients, blood cultures were taken but all remained negative. In section 2.3 we added the information that in the pool of single infections, 2 patients were positive for norovirus, 1 for rotavirus and 1 patient suffered from pyelonephriitis and enterobacter was identified within the urine sample. I the group co-infections, 1 co-infection was with HHV6 and Salmonella typhi. That means, that the group “no pathogen” sums up all cases in which neither a virus within NPA or a virus or bacteria within stool and/or urine and/or blood culture was detected.

Ethics: Technically, children can give assent, no formal legal consent, so suggest to rephrase.

Author's Response:  We added this.

Editing Comments: suggest to check if each legend needs information on the abbreviations used (you could check some other papers in the journal);

Author's Response: For table 2 we added information on the abbreviations used.

Do we need the ‘bullet’ points in table 1 ?

Author's Response: We formatted the table according to the Reviewers suggestions.

We hope to have adequately addressed all issues raised by the reviewers and do hope our revised manuscript is now acceptable for publication in “Pathogens”.

Yours sincerely,

Prof. Dr. Tobias Tenenbaum

Round 2

Reviewer 1 Report

I read carefully the revised manuscript. I saw that part of the recommendations are performed by the authors. However, I think the authors must to perform and other recommendations (indicated by me earlier) to cover their manuscript high requirements of a scientific journal "Pathogens" (IF: 3.492). In this regard, I have the following important recommendations.

1. Title: In connection with the limitations of your research, I highly recommend to change the title of your manuscript with the following new title "General Characteristics of Children with Single-and Co-Infections and Febrile Seizures with a Main Focus on Respiratory Pathogens: Preliminary Results".

2. Introduction: I highly recommend to this section to add 3-4 sentences for "Fever of Unknown Origin (FUO)" among children. This is very important medical problem, for which completely convinced I recommend to write / add information. In this regard, I highly recommend to add the following scientific publications to this section (and add to section "References"):  

  • Chien YL, Huang FL, Huang CM, Chen PY. Clinical approach to fever of unknown origin in children. J Microbiol Immunol Infect. 2017; 50(6): 893-898. [DOI: 10.1016/j.jmii.2015.08.007] [PMID: 27143687].
  • Baymakova M, Plochev K, Dikov I, Popov GT, Mihaylova-Garnizova R, Kovaleva V, et al. Fever of unknown origin in a Bulgarian hospital: Evaluation of 54 cases for a four year-period. J Clin Anal Med. 2016; 7(1): 70-75. [DOI: 10.4328/JCAM.3897].
  • Pijl JP, Kwee TC, Legger GE, Peters HJH, Armbrust W, Schölvinck EH, Glaudemans AWJM. Role of FDG-PET/CT in children with fever of unknown origin. Eur J Nucl Med Mol Imaging. 2020; 47(6): 1596-1604. [DOI: 10.1007/s00259-020-04707-z] [PMID: 32030452].
  • Szymanski AM, Clifford H, Ronis T. Fever of unknown origin: A retrospective review of pediatric patients from an urban, tertiary care center in Washington, DC. World J Pediatr. 2020; 16(2): 177-184. [DOI: 10.1007/s12519-019-00237-3] [PMID: 30888665].

3. Results, Table 1: I highly recommend, to change the term "gender" with a new term "sex."   

4. Results, Table 1: I highly recommend, to add a new column "P-value". For this new column compare "Single infection" and "Co-Infection" (to receive "P-value"). In its current form in Table 1 this is my recommendation is fulfilled only for "Benzodiazepin use". I highly recommend this recommendation to be implemented for everyone else indicators (rows) in Table 1.   

5. Results, Table 2: I think this table in this form has low quality and confuses readers. In this regard, I highly recommend, to add a new column "P-value". For this new column compare "Single infection" and "Co-Infection" (to receive "P-value"). I highly recommend this recommendation to be implemented for everyone indicators (rows) in Table 2.

6. Discussion, Limitations of the Study: I highly recommend to change the following sentence "Results and conclusions should be interpreted in this context." with a new sentence as follows "In this regard, the results and conclusions should be interpreted with caution."   

Author Response

Please find enclosed our revised manuscript “Characteristics of children with single- and co-infections and febrile seizures with a focus on respiratory pathogens: preliminary results” that we would like to submit for publication as a Research-Article in Pathogens as part of the special issue “Respiratory Tract Infections in Children”.

In the newly revised version of our manuscript the additional concerns and suggestions of reviewer 1 have all been addressed and changes are marked up using the “Track Changes” function. We thank all reviewers for their time, their suggestions, thoughts and valuable critique. With the help of this critical input the revised manuscript has significantly improved.

Please find below our point-by-point reply to each comment from Reviewer 1.

Reviewer 1

I read carefully the revised manuscript. I saw that part of the recommendations are performed by the authors. However, I think the authors must to perform and other recommendations (indicated by me earlier) to cover their manuscript high requirements of a scientific journal "Pathogens" (IF: 3.492). In this regard, I have the following important recommendations.

1. Title: In connection with the limitations of your research, I highly recommend to change the title of your manuscript with the following new title "General Characteristics of Children with Single-and Co-Infections and Febrile Seizures with a Main Focus on Respiratory Pathogens: Preliminary Results".

Author's Response: We changed the title to “General characteristics of children with single- and co-infections and febrile seizures with a main focus on respiratory pathogens: Preliminary Results”.

2. Introduction: I highly recommend to this section to add 3-4 sentences for "Fever of Unknown Origin (FUO)" among children. This is very important medical problem, for which completely convinced I recommend to write / add information. In this regard, I highly recommend to add the following scientific publications to this section (and add to section "References"):  

  • Chien YL, Huang FL, Huang CM, Chen PY. Clinical approach to fever of unknown origin in children. J Microbiol Immunol Infect. 2017; 50(6): 893-898. [DOI: 10.1016/j.jmii.2015.08.007] [PMID: 27143687].
  • Baymakova M, Plochev K, Dikov I, Popov GT, Mihaylova-Garnizova R, Kovaleva V, et al. Fever of unknown origin in a Bulgarian hospital: Evaluation of 54 cases for a four year-period. J Clin Anal Med. 2016; 7(1): 70-75. [DOI: 10.4328/JCAM.3897].
  • Pijl JP, Kwee TC, Legger GE, Peters HJH, Armbrust W, Schölvinck EH, Glaudemans AWJM. Role of FDG-PET/CT in children with fever of unknown origin. Eur J Nucl Med Mol Imaging. 2020; 47(6): 1596-1604. [DOI: 10.1007/s00259-020-04707-z] [PMID: 32030452].
  • Szymanski AM, Clifford H, Ronis T. Fever of unknown origin: A retrospective review of pediatric patients from an urban, tertiary care center in Washington, DC. World J Pediatr. 2020; 16(2): 177-184. [DOI: 10.1007/s12519-019-00237-3] [PMID: 30888665].

Author's Response: As suggested by the Reviewer, we now added the mentioned publications and the relevant information to the introduction.

3. Results, Table 1: I highly recommend, to change the term "gender" with a new term "sex."   

Author's Response: According to the reviewer`s recommendation we introduced the term “sex”. We apologize, that we missed to change “gender” and “sex” within the first revision within Table 1 in contrast to other sections of the manuscript, where we introduced “sex” already.

4. Results, Table 1: I highly recommend, to add a new column "P-value". For this new column compare "Single infection" and "Co-Infection" (to receive "P-value"). In its current form in Table 1 this is my recommendation is fulfilled only for "Benzodiazepin use". I highly recommend this recommendation to be implemented for everyone else indicators (rows) in Table 1.   

Author's Response: As suggested by the Reviewer, we added all p values.

5. Results, Table 2: I think this table in this form has low quality and confuses readers. In this regard, I highly recommend, to add a new column "P-value". For this new column compare "Single infection" and "Co-Infection" (to receive "P-value"). I highly recommend this recommendation to be implemented for everyone indicators (rows) in Table 2.

Author's Response: We agree with the Reviewer that all tables should be designed in order to allow the reader easy and clear interpretation. Therefore, we optimized table 2 as requested. For Table 2, we decided to include the p-values in a new column for the comparison of three groups namely: no pathogen -  single infection – co- infection and to add for clarity reasons additional p-values for the comparison of single vs co- infection in the text section 2.4. In total, all relevant p-values are provided, now.

6. Discussion, Limitations of the Study: I highly recommend to change the following sentence "Results and conclusions should be interpreted in this context." with a new sentence as follows "In this regard, the results and conclusions should be interpreted with caution."   

Author's Response: We added “with caution” to the mentioned sentence.

We hope to now have adequately addressed all issues raised by the reviewer and do hope our revised manuscript is acceptable for publication in “Pathogens”.

Yours sincerely,

Prof. Dr. Tobias Tenenbaum